# DiffEdit: Diffusion-based semantic image editing with mask guidance

**Guillaume Couairon, Jakob Verbeek, Holger Schwenk**
Meta AI
{gcouairon,jjverbeek,
schwenk}@meta.com

**Matthieu Cord**
Sorbonne Université, Valeo.ai
matthieu.cord@
sorbonne-universite.fr

## Abstract

Image generation has recently seen tremendous advances, with diffusion models allowing to synthesize convincing images for a large variety of text prompts. In this article, we propose DiffEdit, a method to take advantage of text-conditioned diffusion models for the task of semantic image editing, where the goal is to edit an image based on a text query. Semantic image editing is an extension of image generation, with the additional constraint that the generated image should be as similar as possible to a given input image. Current editing methods based on diffusion models usually require to provide a mask, making the task much easier by treating it as a conditional inpainting task. In contrast, our main contribution is able to automatically generate a mask highlighting regions of the input image that need to be edited, by contrasting predictions of a diffusion model conditioned on different text prompts. Moreover, we rely on latent inference to preserve content in those regions of interest and show excellent synergies with mask-based diffusion. DiffEdit achieves state-of-the-art editing performance on ImageNet. In addition, we evaluate semantic image editing in more challenging settings, using images from the COCO dataset as well as text-based generated images.

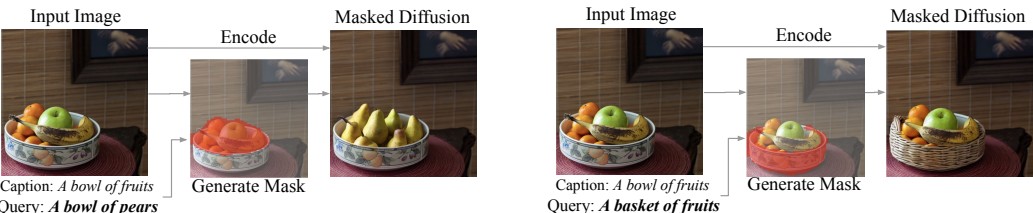

Figure 1: In semantic image editing the goal is to modify an input image based on a textual query, while otherwise leaving the image as close as possible to the original. In our DiffEdit approach, a mask generation module determines which part of the image should be edited, and an encoder infers the latents, to provide inputs to a text-conditional diffusion model which produces the image edit.

## 1 Introduction

The task of semantic image editing consists in modifying an input image in accordance with a textual transformation query. For instance, given an image of a bowl of fruits and the query *"fruits"* → *"pears"*, the aim is to produce a novel image where the fruits have been changed into pears, while keeping the bowl and the background as similar as possible to the input image. The text query can also be a more elaborate description like *"A basket of fruits"*. See the example edits obtained with DiffEdit in Figure 1. Semantic image editing bears strong similarities with image generation and can be viewed as extending text-conditional image generation with an additional constraint: the generated image should be as close as possible to a given input image.

Text-conditional image generation is currently undergoing a revolution, with DALL-E (Ramesh et al., 2021), Cogview (Ding et al., 2021), Make-a-scene (Gafni et al., 2022), Latent Diffusion Models (Rombach et al., 2022), DALL-E 2 (Ramesh et al., 2022) and Imagen (Saharia et al., 2022b),

vastly improving state of the art in modelling wide distributions of images and allowing for unprecedented compositionality of concepts in image generation. Scaling these models is a key to their success. State-of-the art models are now trained on vast amounts of data, which requires large computational resources. Similarly to language models pretrained on web-scale data and adapted in downstreams tasks with prompt engineering, the generative power of these big generative models can be harnessed to solve semantic image editing, avoiding to train specialized architectures (Li et al., 2020a; Wang et al., 2022a), or to use costly instance-based optimization (Crowson et al., 2022; Couairon et al., 2022; Patashnik et al., 2021).

Diffusion models are an especially interesting class of model for image editing because of their iterative denoising process starting from random Gaussian noise. This process can be guided through a variety of techniques, like CLIP guidance (Nichol et al., 2021; Avrahami et al., 2022; Crowson, 2021), and inpainting by copy-pasting pixel values outside a user-given mask (Lugmayr et al., 2022). These previous works, however, lack two crucial properties for semantic image editing: (i) inpainting discards information about the input image that should be used in image editing (e.g. changing a dog into a cat should not modify the animal's color and pose); (ii) a mask must be provided as input to tell the diffusion model what parts of the image should be edited. We believe that while drawing masks is common on image editing tools like Photoshop, language-guided editing offers a more intuitive interface to modify images that requires less effort from users.

Conditioning a diffusion model on an input image can also be done without a mask, e.g. by considering the distance to input image as a loss function (Crowson, 2021; Choi et al., 2021), or by using a noised version of the input image as a starting point for the denoising process as in SDEdit (Meng et al., 2021). However, these editing methods tend to modify the entire image, whereas we aim for localized edits. Furthermore, adding noise to the input image discards important information, both inside the region that should be edited and outside.

To leverage the best of both worlds, we propose DIFFEDIT, an algorithm that leverages a pretrained text-conditional diffusion model for zero-shot semantic image editing, without expensive editing-specific training. DIFFEDIT makes it possible by automatically finding what regions of an input image should be edited given a text query, by contrasting the predictions of a conditional and unconditional diffusion model. We also show how using a reference text describing the input image and similar to the query, can help obtain better masks. Moreover, we demonstrate that using a reverse denoising model, to encode the input image in latent space, rather than simply adding noise to it, allows to better integrate the edited region into the background and produces more subtle and natural edits. See Figure 1 for illustrations. We quantitatively evaluate our approach and compare to prior work using images of the ImageNet and COCO dataset, as well as a set of generated images.

## 2 RELATED WORK

**Semantic image editing.** The field of image editing encompasses many different tasks, from photo colorization and retouching (Shi et al., 2020), to style transfer (Jing et al., 2019), inserting objects in images (Gafni & Wolf, 2020; Brown et al., 2022), image-to-image translation (Zhu et al., 2017; Saharia et al., 2022a), inpainting (Yu et al., 2018), scene graph manipulation (Dhamo et al., 2020), and placing subjects in novel contexts (Ruiz et al., 2022). We focus on semantic image editing, where the instruction to modify an image is given in natural language. Some approaches involve training an end-to-end architecture with a proxy objective before being adapted to editing at inference time, based on GANs (Li et al., 2020b;a; Ma et al., 2018; Alami Mejjati et al., 2018; Mo et al., 2018) or transformers (Wang et al., 2022a; Brown et al., 2022; Issenhuth et al., 2021). Others (Crowson et al., 2022; Couairon et al., 2022; Patashnik et al., 2021; Bar-Tal et al., 2022) rely on optimization of the image itself, or a latent representation of it, to modify an image based on a high-level multimodal objective in an embedding space, typically using CLIP (Radford et al., 2021). These approaches are quite computationnaly intensive, and work best when the optimization is coupled with a powerful generative network. Given a pre-trained generative model such as a GAN, it has also been explored to find directions in the latent space that corresponds to specific semantic edits (Härkönen et al., 2020; Collins et al., 2020; Shen et al., 2020; Shoshan et al., 2021), which then requires GAN inversion to edit real images (Wang et al., 2022c; Zhu et al., 2020; Grechka et al., 2021).

**Image editing with diffusion models.** Because diffusion models iteratively refine an image starting from random noise, they are easily adapted for inpainting when a mask is given as input. Song et al.

(2021) proposed to condition the generation process by copy-pasting pixel values from the reference image at each denoising step. Nichol et al. (2021) use a similar technique by copy-pasting pixels in the estimated final version of the image. Wang et al. (2022b) use DDIM encoding of the input image, and then decode on edited sketches or semantic segmentation maps. The gradient of a CLIP score can also be used to match a given text query inside a mask, as in Paint by Word (Bau et al., 2021), local CLIP-guided diffusion (Crowson, 2021), or blended diffusion (Avrahami et al., 2022). Lugmayr et al. (2022) apply a sequence of noise-denoise operations to better inpaint a specific region. There are also a number of methods that do not require an editing mask. In DiffusionCLIP (Kim & Ye, 2021), the weights of the diffusion model themselves are updated via gradient descent from a CLIP loss with a target text. The high computational cost of fine-tuning a diffusion model for each input image, however, makes it impractical as an interactive image editing tool. In SDEdit (Meng et al., 2021) the image is corrupted with Gaussian noise, and then the diffusion network is used to denoise it. While this method is originally designed to transform sketches to real images and to make pixel-based collages more realistic, we adapt it by denoising the image conditionally to the text query. In ILVR (Choi et al., 2021), the decoding process of diffusion model is guided with the constraint that downsampled versions of the input image and decoded image should stay close. Finally, in recent work concurrent to ours, Hertz et al. (2022) propose to edit images by modifying attention maps during the diffusion process.

## 3 DIFFEDIT FRAMEWORK

In this section, we first give an overview of diffusion models. We then describe our DIFFEDIT approach in detail, and provide a theoretical analysis comparing DIFFEDIT with SDEdit.

### 3.1 BACKGROUND: DIFFUSION MODELS, DDIM AND ENCODING

Denoising diffusion probabilistic models (Ho et al., 2020) is a class of generative models that are trained to invert a diffusion process. For a number of timesteps $T$, the diffusion process gradually adds noise to the input data, until the resulting distribution is (almost) Gaussian. A neural network is then trained to reverse that process, by minimizing the denoising objective

$$\mathcal{L} = \mathbb{E}_{\mathbf{x}_0, t, \boldsymbol{\epsilon}} \|\boldsymbol{\epsilon} - \epsilon_\theta(\mathbf{x}_t, t)\|_2^2, \tag{1}$$

where $\epsilon_\theta$ is the noise estimator which aims to find the noise $\boldsymbol{\epsilon} \sim \mathcal{N}(\mathbf{0}, \mathbf{I})$ that is mixed with an input image $\mathbf{x}_0$ to yield $\mathbf{x}_t = \sqrt{\alpha_t}\mathbf{x}_0 + \sqrt{1 - \alpha_t}\boldsymbol{\epsilon}$. The coefficient $\alpha_t$ defines the level of noise and is a decreasing function of the timestep $t$, with $\alpha_0 = 1$ (no noise) and $\alpha_T \approx 0$ (almost pure noise).

Song et al. (2021) propose to use $\epsilon_\theta$ to generate new images with the *DDIM* algorithm: starting from $\mathbf{x}_T \sim \mathcal{N}(\mathbf{0}, \mathbf{I})$, the following update rule is applied iteratively until step 0:

$$\mathbf{x}_{t-1} = \sqrt{\alpha_{t-1}} \left( \frac{\mathbf{x}_t - \sqrt{1 - \alpha_t}\epsilon_\theta(\mathbf{x}_t, t)}{\sqrt{\alpha_t}} \right) + \sqrt{1 - \alpha_{t-1}}\epsilon_\theta(\mathbf{x}_t, t). \tag{2}$$

The variable $\mathbf{x}$ is updated by taking small steps in the direction of $\epsilon_\theta$. Equation 2 can be written as the neural ODE , taking $\mathbf{u} = \mathbf{x}/\sqrt{\alpha}$ and $\tau = \sqrt{1/\alpha - 1}$:

$$d\mathbf{u} = \epsilon_\theta(\frac{\mathbf{u}}{\sqrt{1 + \tau^2}}, t)d\tau. \tag{3}$$

This allows to view DDIM sampling as an Euler scheme for solving Equation 3 with initial condition $\mathbf{u}(t = T) \sim \mathcal{N}(\mathbf{0}, \alpha_T \mathbf{I})$. This illustrates that we can use fewer sampling steps during inference than the value of $T$ chosen during training, by using a coarser discretization of the ODE. In the remainder of the paper, we parameterize the timestep $t$ to be between 0 and 1, so that $t = 1$ corresponds to $T$ steps of diffusion in the original formulation. As proposed by Song et al. (2021), we can also use this ODE to encode an image $\mathbf{x}_0$ onto a latent variable $\mathbf{x}_r$ for a timestep $r \leq 1$, by using the boundary condition $\mathbf{u}(t = 0) = \mathbf{x}_0$ instead of $\mathbf{u}(t = 1)$, and applying an Euler scheme until timestep $r$. In the remainder of the paper, we refer to this encoding process as *DDIM encoding*, we denote the corresponding function that maps $\mathbf{x}_0$ to $\mathbf{x}_r$ as $E_r$, and refer to the variable $r$ as the *encoding ratio*. Similarly, we note $D_r$ the inverse function that maps $\mathbf{x}_r$ to $\mathbf{x}_0$, which corresponds to regular DDIM decoding. With sufficiently small steps in the Euler scheme, decoding $\mathbf{x}_r$ approximately recovers the original image $\mathbf{x}_0$. This property is particularly interesting in the context of image editing: all the information of the input image $\mathbf{x}_0$ is encoded in $\mathbf{x}_r$, and can be accessed via DDIM sampling.

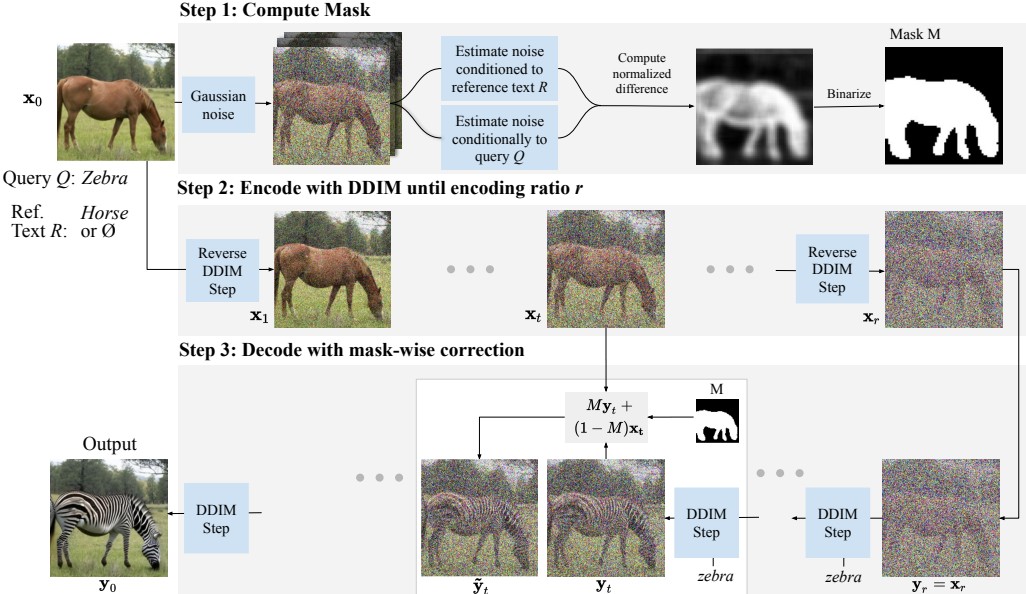

Figure 2: The three steps of DIFFEDIT. **Step 1:** we add noise to the input image, and denoise it: once conditioned on the query text, and once conditioned on a reference text (or unconditionally). We derive a mask based on the difference in the denoising results. **Step 2:** we encode the input image with DDIM, to estimate the latents corresponding to the input image. **Step 3:** we perform DDIM decoding conditioned on the text query, using the inferred mask to replace the background with pixel values coming from the encoding process at the corresponding timestep.

## 3.2 SEMANTIC IMAGE EDITING WITH DIFFEDIT

In many cases, semantic image edits can be restricted to only a part of the image, leaving other parts unchanged. However, the input text query does not explicitly identify this region, and a naive method could allow for edits all over the image, risking to modify the input in areas where it is not needed. To circumvent this, we propose DIFFEDIT, a method to leverage a text-conditioned diffusion model to infer a mask of the region that needs to be edited. Starting from a DDIM encoding of the input image, DIFFEDIT uses the inferred mask to guide the denoising process, minimizing edits outside the region of interest. Figure 2 illustrates the three steps of our approach, which we detail below.

**Step 1: Computing editing mask.** When the denoising an image, a text-conditioned diffusion model will yield different noise estimates given different text conditionings. We can consider *where* the estimates are different, which gives information about what image regions are concerned by the change in conditioning text. For instance, in Figure 2, the noise estimates conditioned to the query *zebra* and reference text *horse*[1] are different on the body of the animal, where they will tend to decode different colors and textures depending on the conditioning. For the background, on the other hand, there is little change in the noise estimates. The difference between the noise estimates can thus be used to infer a mask that identifies what parts on the image need to be changed to match the query. In our algorithm, we use a Gaussian noise with strength 50% (see analysis in Appendix A.1), remove extreme values in noise predictions and stabilize the effect by averaging spatial differences over a set of $n$ input noises, with $n = 10$ in our default configuration. The result is then rescaled to the range $[0, 1]$, and binarized with a threshold, which we set to 0.5 by default. The masks generally somewhat overshoot the region that requires editing, this is beneficial as it allows it to be smoothly embedded in it's context, see examples in Section 4 and Section A.5.

**Step 2: Encoding.** We encode the input image $x_0$ in the implicit latent space at timestep $r$ with the DDIM encoding function $E_r$. This is done with the unconditional model, i.e. using conditioning text $\emptyset$, so no text input is used for this step.

---

[1]We can also use an empty reference text, which we denote as $Q = \emptyset$.

**Step 3: Decoding with mask guidance.** After obtaining the latent $\mathbf{x}_r$, we decode it with our diffusion model conditioned on the editing text query $Q$, e.g. *zebra* in the example of Figure 2. We use our mask $M$ to guide this diffusion process. Outside the mask $M$, the edited image should in principle be the same as the input image. We guide the diffusion model by replacing pixel values outside the mask with the latents $\mathbf{x}_t$ inferred with DDIM encoding, which will naturally map back to the original pixels through decoding, unlike when using a noised version of $\mathbf{x}_0$ as typically done (Meng et al., 2021; Song et al., 2021). The mask-guided DDIM update can be written as $\tilde{\mathbf{y}}_t = M\mathbf{y}_t + (1-M)\mathbf{x}_t$, where $\mathbf{y}_t$ is computed from $\mathbf{y}_{t-dt}$ with Eq. 2, and $\mathbf{x}_t$ is the corresponding DDIM encoded latent.

The encoding ratio $r$ determines the strength of the edit: larger values of $r$ allow for stronger edits that allow to better match the text query, at the cost of more deviation from the input image which might not be needed. We evaluate the impact of this parameter in our experiments. We illustrate the effect of the encoding ratio in Appendix A.5.

## 3.3 THEORETICAL ANALYSIS

In DIFFEDIT, we use DDIM encoding to encode images before doing the actual editing step. In this section, we give theoretical insight on why this component yields better editing results than adding random noise as in SDEdit (Meng et al., 2021). With $\mathbf{x}_r$ being the encoded version of $\mathbf{x}_0$, using DDIM decoding on $\mathbf{x}_r$ unconditionally would give back the original image $\mathbf{x}_0$. In DIFFEDIT, we use DDIM decoding conditioned on the text query $Q$, but there is still a strong bias to stay close to the original image. This is because the unconditional and conditional noise estimator networks $\epsilon_\theta$ and $\epsilon_\theta(\cdot, Q)$ often produce similar estimates, yielding similar decoding behavior when initialized with the same starting point $\mathbf{x}_r$. This means that the edited image will have a small distance w.r.t. the input image, a property critical in the context of image editing. We capture this phenomenon with the proposition below, where we compare the DDIM encoder $E_r(\mathbf{x}_0)$ to the SDEdit encoder $G_r(\mathbf{x}_0, \epsilon) := \sqrt{\alpha_r}\mathbf{x}_0 + \sqrt{1 - \alpha_r}\epsilon$, which simply adds noise to the image $\mathbf{x}_0$.

**Proposition 1.** *Let $\mathcal{X} = \mathbb{R}^d$ be the space of input images, $p_D$ be the data distribution of couples $(\mathbf{x}_0, Q)$ where $\mathbf{x}_0 \in \mathcal{X}$ and $Q$ a textual query to edit that image. Suppose that $\|\epsilon_\theta(\mathbf{x}_t, Q, t)\|_2 \leq C$ for all $x \in \mathcal{X}$, $t \in [0, 1]$, that $\epsilon_\theta(\cdot, \emptyset, t)$ is $K_1$-Lipschitz for all $t$, and let $K_2 = \mathbb{E}_{(\mathbf{x}_0, Q) \in p_D} \max_{t \in [0,1]} \|\epsilon_\theta(\mathbf{x}, Q, t) - \epsilon_\theta(\mathbf{x}, \emptyset, t)\|$. Then, for all encoding ratios $0 \leq r \leq 1$, we have the two following bounds:*

$$\mathbb{E}_{\substack{(\mathbf{x}_0, Q) \sim p_D \\ \epsilon \sim \mathcal{N}(0,1)}} \|\mathbf{x}_0 - D_r(G_r(\mathbf{x}_0, \epsilon), Q)\|_2 \leq (C + 1)\tau, \tag{4}$$

$$\mathbb{E}_{(\mathbf{x}_0, Q) \sim p_D} \|\mathbf{x}_0 - D_r(E_r(\mathbf{x}_0), Q)\|_2 \leq \frac{K_2 \tau}{\sqrt{\tau^2 + 1}} \left(\tau + \sqrt{\tau^2 + 1}\right)^{K_1}, \tag{5}$$

*where $\tau = \sqrt{1/\alpha_r - 1}$ increases with the encoding ratio $r$: $\tau(r=0) = 0$ and $\lim_{r \to 1} \tau = +\infty$.*

We provide the proof in Appendix B. The first bound is associated with SDEdit, and is an extension of a bound proven in the original paper. The second bound we contribute is associated with DIFFEDIT. It is tighter than the first bound below a certain encoding ratio, see Figure 3. We empirically estimated the parameters $K_1, K_2$ and $C$ with the diffusion models that we are using. While asymptotic behavior of the second bound is worse than the first with $K_1 > 1$, it is the very small value of $K_2$ that gives a tighter bound.

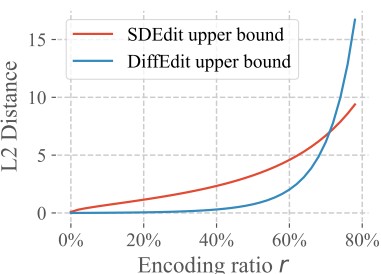

This supports our argument from above: because the unconditional and text-conditional noise estimates generally give close results —$K_2$ being a measure of the average difference— the Euler scheme with $\epsilon_\theta(\cdot, Q, \cdot)$ gives a sequence of intermediate latents $\mathbf{y}_r, ..., \mathbf{y}_0$ that stays close to the trajectory $x_r, ..., D_r(x_r) \approx \mathbf{x}_0$ mapping back $\mathbf{x}_r$ to $\mathbf{x}_0$. While these upper bounds do not guarantee that DDIM encoding yields smaller edits than SDEdit, experimentally we find that it is indeed the case.

Figure 3: Illustration of the bounds from Proposition 1, with estimated parameters $C = 1$, $K_2 = 0.02$, and $K_1 = 3$.

# 4 EXPERIMENTS

In this section, we describe our experimental setup, followed by qualitative and quantitative results.

## 4.1 EXPERIMENTAL SETUP

**Datasets.** We perform experiments on three datasets. First, on *ImageNet* (Deng et al., 2009) we follow the evaluation protocol of FlexIT (Couairon et al., 2022). Given an image belonging to one class, the goal is to edit it so that it will depict an object of another class as indicated by the query. Given the nature of the ImageNet dataset, edits often concern the main object in the scene. Second, we consider editing images generated by *Imagen* (Saharia et al., 2022b) based on structured text prompts, in order to evaluate edits that involve changing the background, replacing secondary objects, or changing object properties. Third, we consider edits based on images and queries from the *COCO* (Lin et al., 2014) dataset to evaluate edits based on more complex text prompts.

**Diffusion models.** In our experiments we use latent diffusion models (Rombach et al., 2022). We use the class-conditional model trained on ImageNet at resolution $256 \times 256$, as well as the 890M parameter text-conditional model trained on LAION-5B (Schuhmann et al., 2021), known as *Stable Diffusion*, at $512 \times 512$ resolution.[2] Since these models operate in a VQGAN latent spaces (Esser et al., 2021), the resolution of our masks is $32 \times 32$ (ImageNet) or $64 \times 64$ (Imagen and COCO). We use 50 steps in DDIM sampling with a fixed schedule, and the encoding ratio parameter further decreases the number of updates used for our edits. This allows to edit images in ∼10 seconds on a single Quadro GP100 GPU. We also use classifier-free guidance (Ho & Salimans, 2022) with the recommended values: 5 on ImageNet, 7.5 for Stable Diffusion. For more details see Section A.2.

**Comparison to other methods.** We use SDEdit (Meng et al., 2021) as our main point of comparison, since we can use the same diffusion model as for DIFFEDIT. We also compare to FlexIT (Couairon et al., 2022), a mask-free, optimization-based editing method based on VQGAN and CLIP. On ImageNet, we evaluate ILVR (Choi et al., 2021) which uses another diffusion model trained on ImageNet (Dhariwal & Nichol, 2021). Finally, on COCO and Imagen images, we compare to the concurrent work of Hertz et al. (2022). [3]

**Evaluation.** In semantic image editing, we have to satisfy the two contradictory objectives of (i) matching the text query and (ii) staying close to the input image. For a given editing method, better matching the text query comes at the cost of increased distance to the input image. Different editing methods often have a parameter that allows to control the editing strength: varying its value allows to get different operating points, forming a trade-off curve between the two objectives aforementioned. Therefore, we evaluate editing methods by comparing their trade-off curves. For diffusion-based methods, we use the encoding ratio to control the trade-off.

## 4.2 EXPERIMENTS ON IMAGENET

On ImageNet, we follow the evaluation protocol of Couairon et al. (2022), with the associated metrics: the LPIPS perceptual distance (Zhang et al., 2018) measures the distance with the input image, and the CSFID, which is a class-conditional FID metric (Heusel et al., 2017) measuring both image realism and consistency w.r.t. the transformation prompt. For both metrics, lower values indicate better edits. For more details see Couairon et al. (2022).

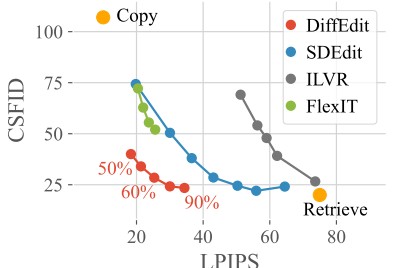

Figure 4: Comparison on ImageNet data of DIFFEDIT with other Image Editing methods. For DIFFEDIT we annotate the different operating points with the corresponding encoding ratios.

We compare DIFFEDIT to other semantic editing methods from the literature in terms of CSFID-LPIPS trade-off. Stronger edits improve (lower) the CSFID score as the edited images better adhere to the text query, but the resulting images tend to deviate more from the input image, leading to worse (increased) LPIPS distances.

---

[2]Available at `https://huggingface.co/CompVis/stable-diffusion`.

[3]As there is no official implementation available at the time of writing, we used the unofficial implementation adapted for Stable Diffusion from `https://github.com/bloc97/CrossAttentionControl`.

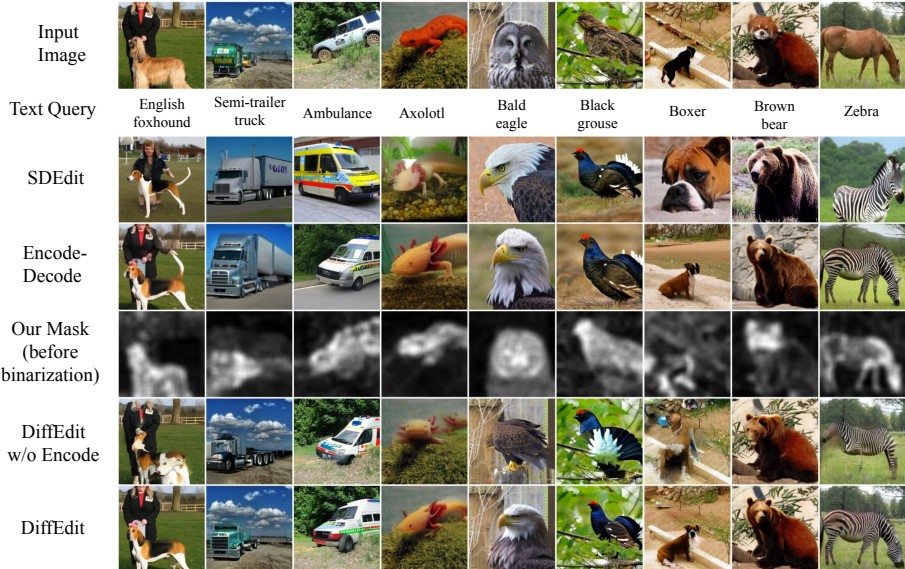

Figure 5: Edits obtained on ImageNet with DIFFEDIT and ablated models. Encode-Decode is DIFFEDIT without masking, and SDEdit is obtained when not using masking nor encoding. When not using masking (SDEdit and Encode-Decode) we observe undesired edits to the background, see e.g. the sky in the second column. When not using DDIM encoding (SDEdit and DiffEdit w/o Encode), appearance information from the input —such as pose— is lost, see last two columns.

The results in Figure 4 indicate that DIFFEDIT obtains the best trade-offs among the different methods. For fair comparison with previous methods, here we do not leverage the label of the input image and use the empty text as reference when inferring the editing mask. The *Copy* and *Retrieve* baselines are two opposite cases where we have best possible LPIPS distance —zero, by copying the input image— and best possible transformation score by discarding the input image and replacing it with a real image from the target class from the ImageNet dataset. DIFFEDIT, as well as the diffusion-based SDEdit and ILVR, are able to obtain CSFID values comparable to that of the retrieval baseline. Among the diffusion-based methods, our DIFFEDIT obtains comparable CSFID values at significantly better LPIPS scores. For FlexIT, the CSFID best value is significantly worse, indicating it is not able to produce both strong and realistic edits. Using more optimization steps does not solve this issue, as the distance to the input image is part of the loss it minimizes.

**Ablation experiments.** We ablate the two core components of DIFFEDIT, mask inference and DDIM encoding, to measure their relative contributions in terms of CSFID-LPIPS trade-off. If we do not use either of these components our method reverts to SDEdit (Meng et al., 2021). The results in Figure 6, left panel, show that adding DDIM encoding (Encode-Decode) and the masking (DiffEdit w/o Encode) separately both improve the trade-off and reduce the average editing distance w.r.t. the input image compared to SDEdit. Moreover, combining these two elements into DIFFEDIT gives an even better trade-off, showing their complementarity: masking better preserves the background, while DDIM encoding retains visual appearance of the input inside the mask. See Figure 5 for qualitative examples of these ablations, along with the inferred masks.

The right panel of Figure 6 shows DIFFEDIT with different mask binarization thresholds. Compared to our default value of 0.5, a lower threshold of 0.25 results in larger masks (more image modifications) and worse CSFID-LPIPS tradeoff. A higher threshold of 0.75 results in masks that are too restrictive: the CSFID score stagnates around 40, even at large encoding ratios.

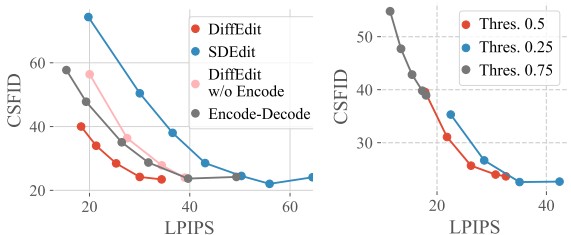

Figure 6: Ablations on ImageNet. Left: effect of masking and encoding component. Right: DIFFEDIT with different mask thresholds; with 0.5 our default setting.

Finally, our mask guidance operator $\tilde{\mathbf{y}}_t = M\mathbf{y}_t + (1 - M)\mathbf{x}_t$ provides a better trade-off than the operator used in GLIDE (Nichol et al., 2021), which interpolates $\mathbf{y}_t$ with a mask-corrected version of the predicted denoised image $\hat{\mathbf{y}}_0$. With encoding ratio 80%, both operators produce edits with a LPIPS score of 30.5, but the GLIDE version yields a CSFID of 26.4 compared to 23.6 for ours.

### 4.3 EXPERIMENTS ON IMAGES GENERATED BY IMAGEN

In our second set of experiments we evaluate edits that involve changes in background, replacing secondary objects, and editing object properties. We find that images generated by Imagen (Saharia et al., 2022b) offer a well suited testbed for this purpose. Indeed, the authors tested the compositional abilities of Imagen with templated prompts of the form: *"{A photo of a | An oil painting of a} {fuzzy panda | British shorthair cat | Persian cat | Shiba Inu dog | raccoon} {wearing a cowboy hat and | wearing sunglasses and} {red shirt | black jacket} {playing a guitar | riding a bike | skateboarding} {in a garden | on a beach | on top of a mountain}"*, resulting in 300 prompts.

We use the generated images as input and ask to change the prompt to another prompt for which one of these elements is changed. Since we cannot use the CSFID metric as for ImageNet, as images do not carry a single class label, we use FID to measure image realism, and CLIP-Score (Hessel et al., 2021) to measure the alignment of the query and output image. These two scores have become the standard in evaluating text-conditional image generation (Saharia et al., 2022b).

Figure 7 displays the CLIP-LPIPS and FID-CLIP trade-offs. DIFFEDIT provides more accurate edits than SDEdit, FlexIT, and Cross Attention Control, by combining inferred masks with DDIM encoding. Two versions of DiffEdit are shown, which differ by how the mask is computed: they correspond to (i) using the original caption as reference text (labelled *w/ ref. text*) or (ii) using the empty text $\emptyset$ (labelled *w/o ref. text*).

Computing the mask with the original caption as reference text yields the best overall trade-off. Leveraging the original caption yields better CLIP and FID scores. Figure 8 illustrates the difference in the masks obtained with and without reference text for two examples. The reference text allows to ignore parts of the image that are described both by the query and reference text (e.g. the fruits), because in both cases the network uses the common text on the corresponding image region to estimate the noise. On the contrary, parts where the query and reference text disagree, e.g. *"bowl"* vs. *"basket"*, will have different noise estimates.

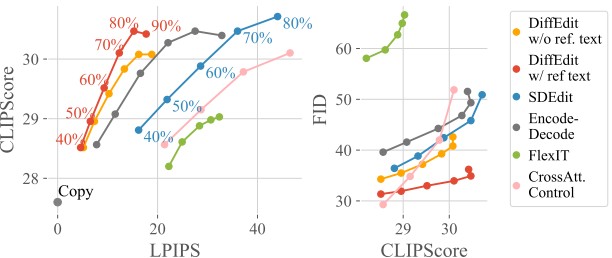

Figure 7: Editing trade-offs on Imagen images.

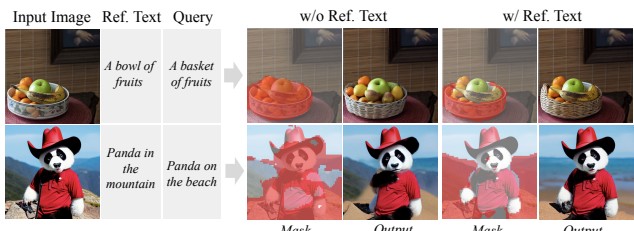

Figure 8: Masks and edits obtained with and without reference text in the mask computation algorithm.

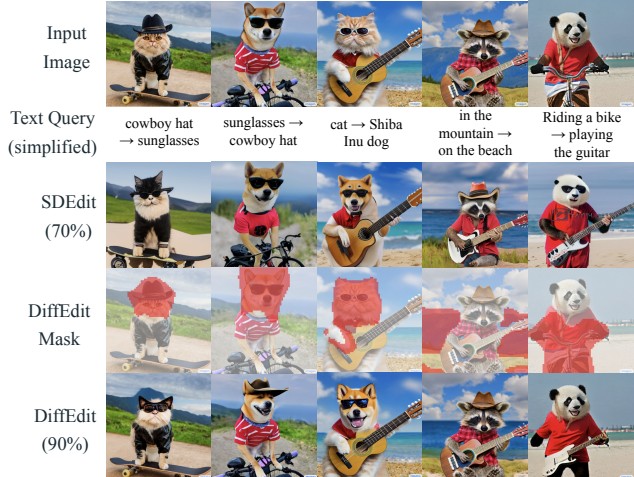

Figure 9: Edits on Imagen dataset. We use encoding ratio of 90% for DiffEdit and 70% for SDEdit for fair comparison: both methods have similar CLIPScore, for larger encoding ratios SDEdit drastically change the input.

Qualitative transformation examples are shown in Figure 9, where the masks are inferred by contrasting the caption and query texts.

## 4.4    EXPERIMENTS ON COCO

To evaluate semantic image editing with more complex prompts, we use images and captions from the COCO dataset Lin et al. (2014). To this end, we leverage the annotations provided by Hu et al. (2019), which associate images from the COCO validation set with other COCO captions that are similar to the original ones, but in contradiction with the given image. This makes these annotations particularly interesting as queries for semantic image editing, as they can often be satisfied by editing only a part of the input image, see Figure 15 in the supplementary material for examples. Similar to our evaluation for Imagen images, here we evaluate edits in terms of CLIPScore, FID and LPIPS.

The results in Figure 10 show that the CLIP-LPIPS trade-off of DIFFEDIT is the best, but that it reaches lower maximum CLIP score than SDEdit. The FID scores are similar to SDEdit, but significantly improves upon the Encode-Decode ablation, which does not use a mask.

Moreover, in contrast to results on the Imagen data, leveraging the original image caption does not change the CLIP-LPIPS and FID-CLIP trade-offs. We find that the caption often describes the input image differently compared to the query text, making it more difficult to identify which part of the image needs to be edited. We verify this hypothesis in Section A.3 by filtering the dataset according to the edit distance between the caption and edit query. When the caption and edit query are similar, leveraging the image caption boosts CLIP scores by 0.25 points, a similar improvement as seen on the Imagen data.

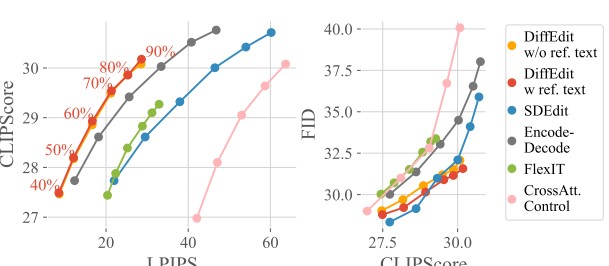

Figure 10:  Quantitative evaluation on COCO.

Qualitative examples are shown in Figure 11. The first column illustrates the benefit of DDIM encoding: we are able to correctly maintain properties of the object inside the mask, such as clothes' color. The three last columns illustrate how contrasting different pairs of reference and query text allows to select different objects in the input image to perform different edits. See Section A.5 for more examples.

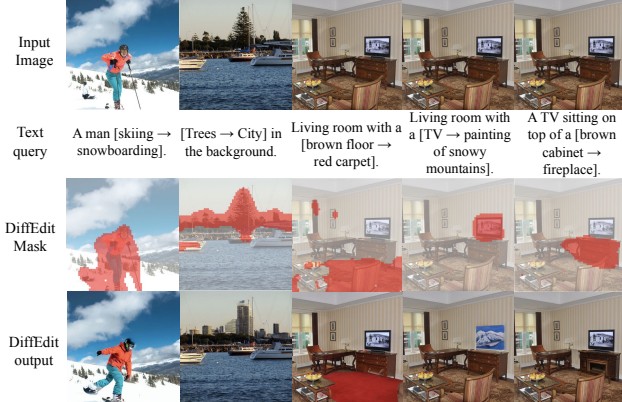

Figure 11: Examples edits on COCO images.

## 5    CONCLUSION

We introduced DIFFEDIT, a novel algorithm for semantic image editing based on diffusion models. Given a textual query, using the diffusion model, DIFFEDIT infers the relevant regions to be edited rather than requiring a user generated mask. Furthermore, in contrast to other diffusion-based methods, we initialize the generation process with a DDIM encoding of the input. We provide theoretical analysis that motivates this choice, and show experimentally that this approach conserves more appearance information from the input image, leading to lighter edits. Quantitative and qualitative evaluations on ImageNet, COCO, and images generated by Imagen, show that our approach leads excellent edits, improving over previous approaches. Although DIFFEDIT works better with a reference text describing the input image, we believe this additional information can be inferred from input image and target caption, which we leave for future work.

## 6 ETHICS STATEMENT

Image editing raises several ethical challenges that we wish to discuss here. First, as image editing is closely related to image generation, it inherits known concerns. Open-source diffusion models are trained on large amounts of web-scraped data like LAION, and inherit their biases. In particular, it was shown that LAION contains inappropriate content (violence, hate, pornography), along with racist and sexist stereotypes. Furthermore it was found that diffusion models trained on LAION, such as Imagen, can exhibit social and cultural bias. Therefore, the use of such models can raise ethical issues, whether the text prompt is intentionnally harmful or not. Because image editing is usually performed on real images, there are additionnal ethical challenges, such as potential skin tone change when editing a person or re-inforcing harmful social stereotypes. We believe that open-sourcing editing algorithms in a research context contributes to a better understanding of such problems, and can aid the community in efforts to mitigate them in the future. Furthermore, image editing tools could be used with harmful intent such as harrassement or propagating fake news. This use, known as deep fakes, has been largely discussed in previous work, e.g. in Etienne (2021). To mitigate potential misuse, the Stable Diffusion model is released under a license focused on ethical and legal use, stating explicitly that users "must not distribute harmful, offensive, dehumanizing content or otherwise harmful representations of people or their environments, cultures, religions, etc. produced with the model weights".

Our editing benchmark based on the COCO dataset also has some limitations. COCO has a predominant western cultural bias, and we are therefore evaluating transformations on a small subset of images mostly associated with western culture. Finding relevant transformation prompts for an image is challenging: while we found it relevant to leverage existing annotations based on COCO, we believe that evaluating image editing models on a less culturally biased dataset is needed.

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

# DIFFEDIT: Diffusion-based semantic image editing with mask generation
# Supplementary Material

In this supplementary material we provide more details on the experiments and methods presented in the main paper. In Section A we provide additional experimental results, including assessment of the impact of the strength of classifier-free guidance, the impact of using reference texts describing the input image, an illustration of the effect of the encoding ratio, more qualitative editing examples, as well as a number of example images and associated reference and query texts on COCO. In Section B we provide proofs that support Proposition 1 in the main paper.

## A ADDITIONAL EXPERIMENTAL RESULTS

### A.1 ANALYSIS OF NOISE USED TO COMPUTE THE MASK

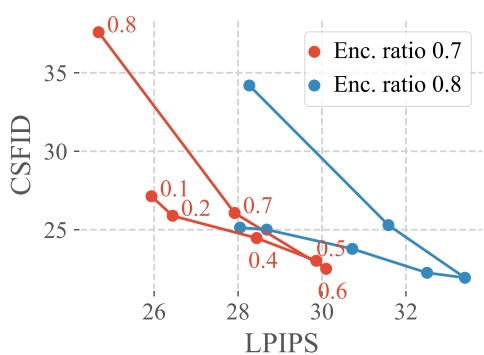

In step one our method an editing mask is inferred by contrasting noise estimations on a *noised* version of the input image, see Section 3.2. In this section, we study the impact of the level of noise added to the input image, by varying its value between 0.1 and 0.8, where 0 corresponds to using the initial image as input, and 1 to replacing the input image with random Gaussian noise. We evaluate the obtained operating points on ImageNet with the CSFID and LPIPS metrics when using a encoding ratios of 0.7 and 0.8 for DDIM encoding and masked-guided denoising in steps two and three of DIFFEDIT. From the results in Figure 12, we find that best results are obtained for moderate values of noise addition of 0.6 and below. Indeed, with too much noise added to the

Figure 12: Impact of the noise added to input image when computing the mask, for a encoding ratios of 0.7 or 0.8 on ImageNet.

input image, it is difficult to correctly identify visual elements in the input image. We use a value of 0.5 in all our experiments.

### A.2 CLASSIFIER-FREE GUIDANCE

In diffusion models, the noise estimator $\epsilon_\theta$ can be conditioned on text describing the image, which provides a signal to guide the noise estimation process (Nichol et al., 2021; Saharia et al., 2022b). Ho & Salimans (2022) introduced classifier-free guidance, a technique to greatly improve generation quality and image-text alignment in text-conditional diffusion models. It consists in training both a conditional and unconditional model by dropping the conditioning text at train time with fixed probability, e.g. 10%. Then, after training, at each step $t$ during decoding, the noise estimation $\epsilon_\theta(\mathbf{x}_t, Q, t)$ is extrapolated by using the unconditional noise estimation $\epsilon_\theta(\mathbf{x}_t, \emptyset, t)$ as origin. Formally, the noise that is used is

$$\epsilon = \epsilon_\theta(\mathbf{x}_t, \emptyset, t) + \lambda(\epsilon_\theta(\mathbf{x}_t, Q, t) - \epsilon_\theta(\mathbf{x}_t, \emptyset, t)),$$

Figure 13: Ablation on the value of the classifier-free guidance parameter. On ImageNet, we find that a value of at least 3 must be used to get good results. We chose to use 5, which is the default value recommended for generation.

where $\lambda$ is the classifier-free guidance parameter. We study the influence of this parameter on DIFFEDIT in Figure 13, finding that similarly to generation, a value above 3 yields the best results. Without classifier-free guidance, the obtained trade-off is not competitive. For our experiments we use a default guidance value of $\lambda = 5$.

## A.3 EXPERIMENTS ON COCO FILTERING

Here, we investigate why there is little difference between using or not the reference text to compute the mask on our COCO queries. In Figure 15 we show several editing queries on the COCO dataset taken from the BISON dataset (Hu et al., 2019). Generally, the text query describes a scene similar to the one in the input image, and it is possible to match the text query by editing only a fraction of the input image.

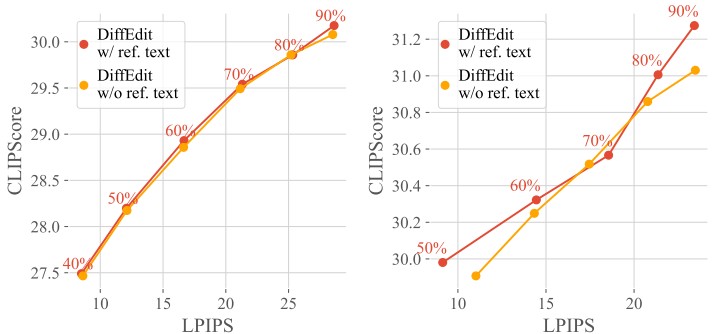

Figure 14: Results on COCO: unfiltered (left) and filtered (right). While having a small impact overall, for the filtered set using the reference text is beneficial, especially at high encoding ratios, e.g. 90%.

However, we find that while queries have been built to be close to a caption of the input image, most of the time the query is not well aligned with the caption. We create a filtered version of this dataset, for which queries are structurally similar to the caption, i.e. where only a few words are changed, but the grammatical structure stays the same. We use the filtering criterion that the total number of words inserted/deleted/replaced must not exceed 25% of the total number of words in the original caption, resulting in a total of 272 queries out of 50k original queries. In Figure 14 we compare results with and without filtering, and observe that for the images with small caption edits the gain of DIFFEDIT (w/ ref. text) compared to *Encode-Decode* is somewhat larger than on the unfiltered dataset. Moreover, using the original caption as reference text to compute the mask gives higher CLIPScore, especially at high encoding ratio. This illustrates that a well chosen reference text helps to generate better editing masks.

| Dataset | COCO | COCO | COCO |
|---|---|---|---|

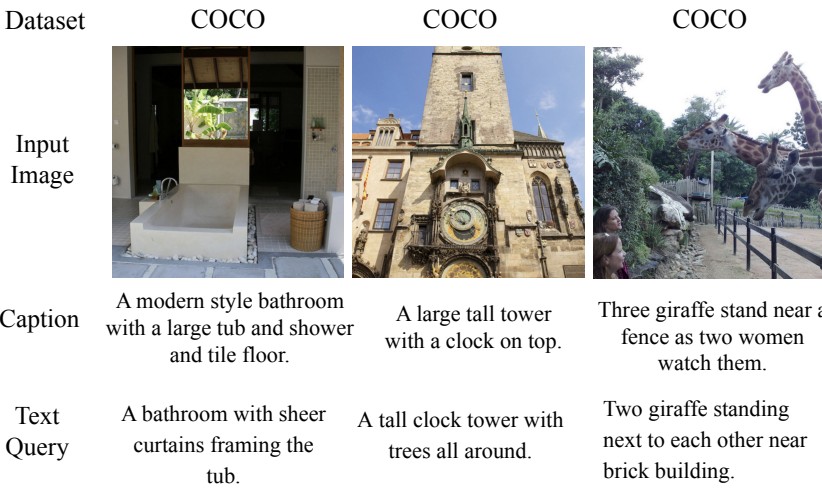

| Input Image | | | |
|---|---|---|---|
| Caption | A modern style bathroom with a large tub and shower and tile floor. | A large tall tower with a clock on top. | Three giraffe stand near a fence as two women watch them. |
| Text Query | A bathroom with sheer curtains framing the tub. | A tall clock tower with trees all around. | Two giraffe standing next to each other near brick building. |

Figure 15: Editing queries on the COCO dataset.

## A.4 VISUALISATION OF THE IMPACT OF ENCODING RATIO

We show visual results for ablations of our two main components, mask inference and DDIM encoding, in Figure 16. The resulting methods are SDEdit, Encode-Decode, DIFFEDIT w/o Encoding, and DIFFEDIT. We demonstrate the qualitative behavior of these different methods, at varying encoding ratios between 30% and 80%. Compared to SDEdit, Encode-Decode allows to better match the query with less modifications of the main object and the background, especially at 60% − 70%.

Mask inference allows to maintain exactly the background. Using DDIM inference on top of mask-based decoding allows to better retain of the content inside the mask, especially at 70% and 80%, c.f. row 3 vs. 4.

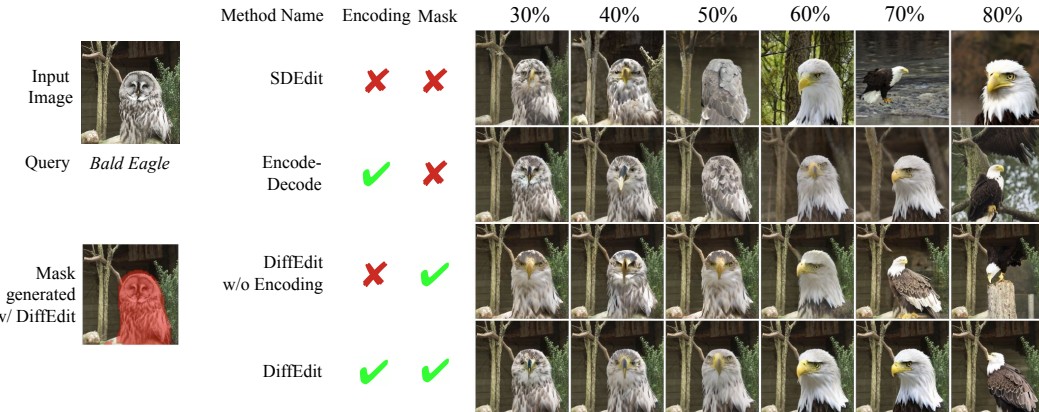

Figure 16: Qualitative ablations of the mask and encoding components, using different encoding ratios from 30% to 80%.

## A.5 ADDITIONAL VISUALIZATIONS AND QUALITATIVE RESULTS

Figure 17 illustrates editing examples on Imagen, in comparison with other mask-free editing methods. DIFFEDIT generally performs more targeted and accurate edits, leaving more of the original image in tact where possible. Consider for example the first column of Figure 17, where DIFFEDIT leaves the guitar as it, while other methods make unnecessary and unrealistic changes to the guitar.

Additional qualitative examples on COCO images are shown in Figure 19.

Figure 20 shows several failure cases of semantic image editing with DIFFEDIT. Some failure modes are inherited from the generative model itself: models trained on web-scrapped image-text data are known to struggle with understanding spatial positions in images, spatial reasoning, and counting (Ramesh et al., 2021). Others are specific to our mask-based method, like the difficulty to insert objects, because the mask often seeks an "anchor" visual element to insert an object, see first column.

## A.6 DETAILS ON COMPARISONS WITH OTHER METHODS

On the COCO and Imagen datasets, we do not compare with ILVR, since it cannot be used within the latent diffusion framework: the method needs image downsampling and upsampling, which does not work well with the latent spaces used in latent diffusion. Even adapted with a diffusion model without latent spaces like Imagen, we do not expect the CLIP-LPIPS trade-off to be favorable for this method, given the high editing distance obtained on ImageNet. Instead, we compare against Cross-Attention Control Hertz et al. (2022), a recent method for text-driven image editing based on the unreleased Imagen diffusion model. The method is very recent and has been adapted to use with Stable Diffusion at https://github.com/bloc97/CrossAttentionControl/. We have performed lightweight hyperparameter search to optimize the CLIP-LPIPS trade-off on a subset of Imagen images. We generally find that this re-implementation, while producing edited images structurally similar to the input, changes local features more than SDEdit, leading to generally high LPIPS distances, resulting in a CLIP-LPIPS trade-off not competitive with other methods on our COCO and Imagen benchmark. In particular, LPIPS distance are high on the COCO dataset where text query and reference text have a high edit distance on average, whereas Cross-Attention Control was designed to perform well for prompt-to-prompt editing, i.e. the input and target text should almost exactly match. Given that our results are based on the unofficial re-implementation, we caution that they are temporary and we will update them when the official code (or official adaptation for Stable Diffusion) is released.

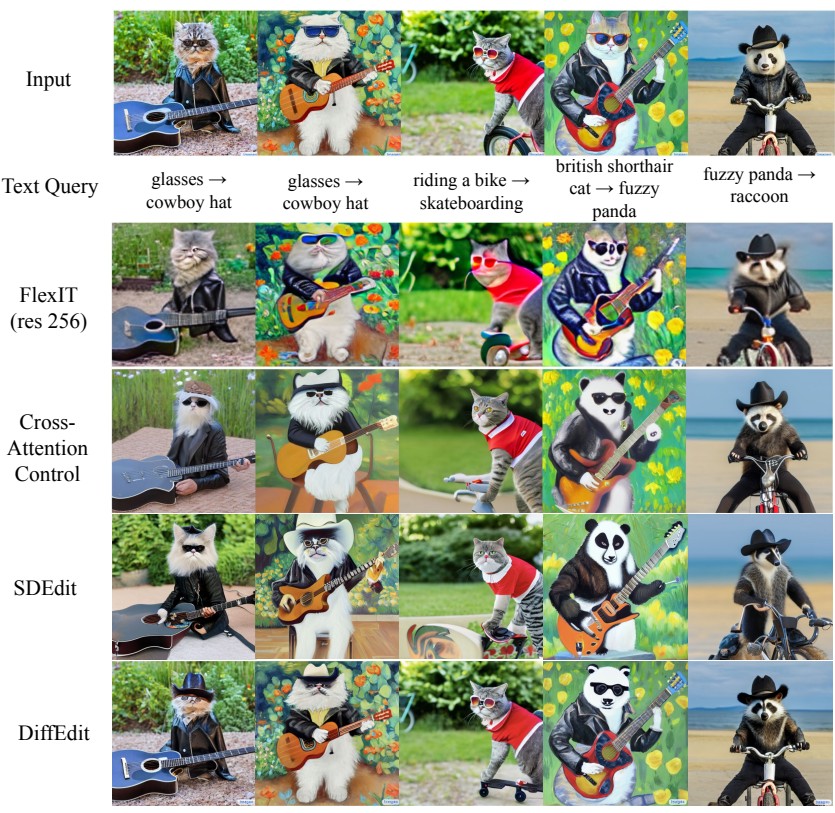

Figure 17: Example edits from Imagen, in comparison with other mask-free editing methods.

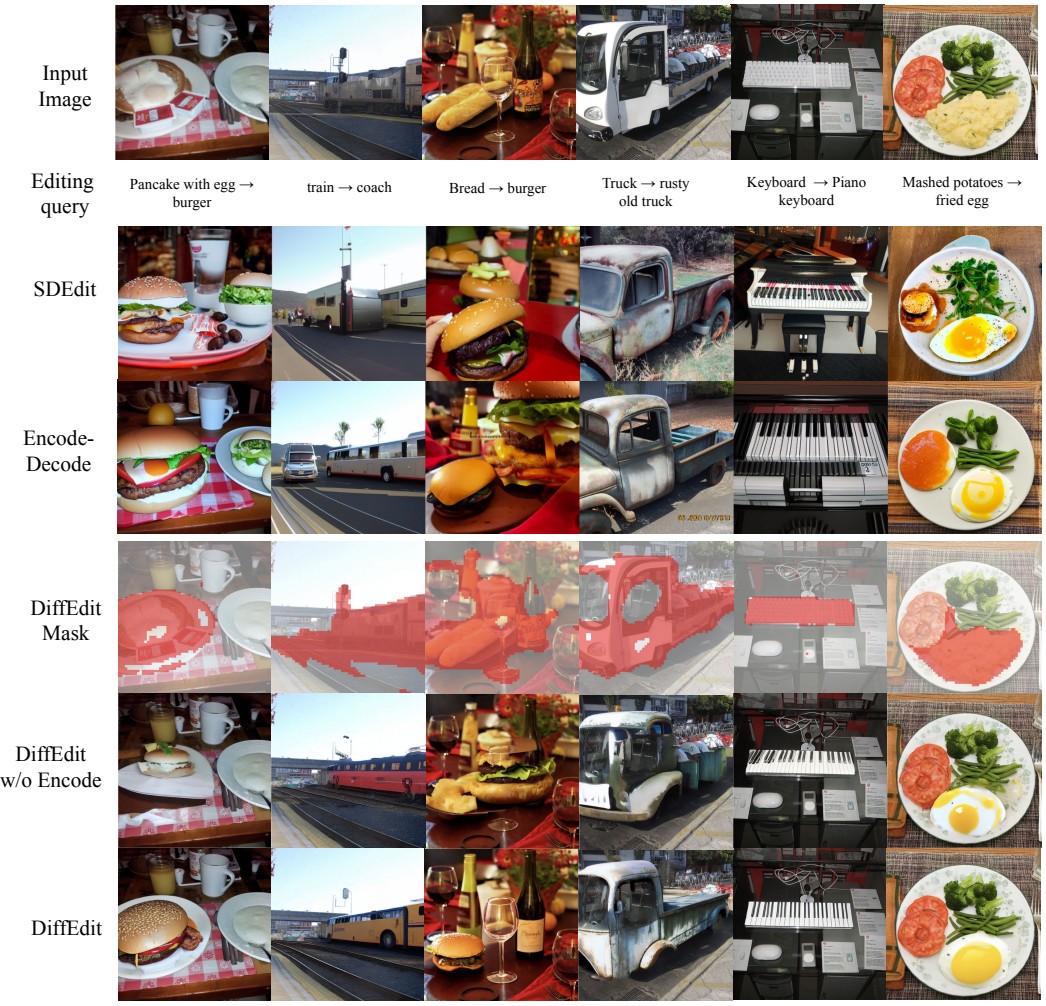

Figure 18: More qualitative examples on COCO. Baseline methods are shown for comparison. The mask is sometimes bigger or smaller that one could expect: In column 3, it is larger, but there are few edits outside the requested *bread → burger* transformation (except for the wine bottle label), which is not the case without DDIM encoding. In column 4, the mask does not cover the interior of the truck, but this does not affect the edit quality.

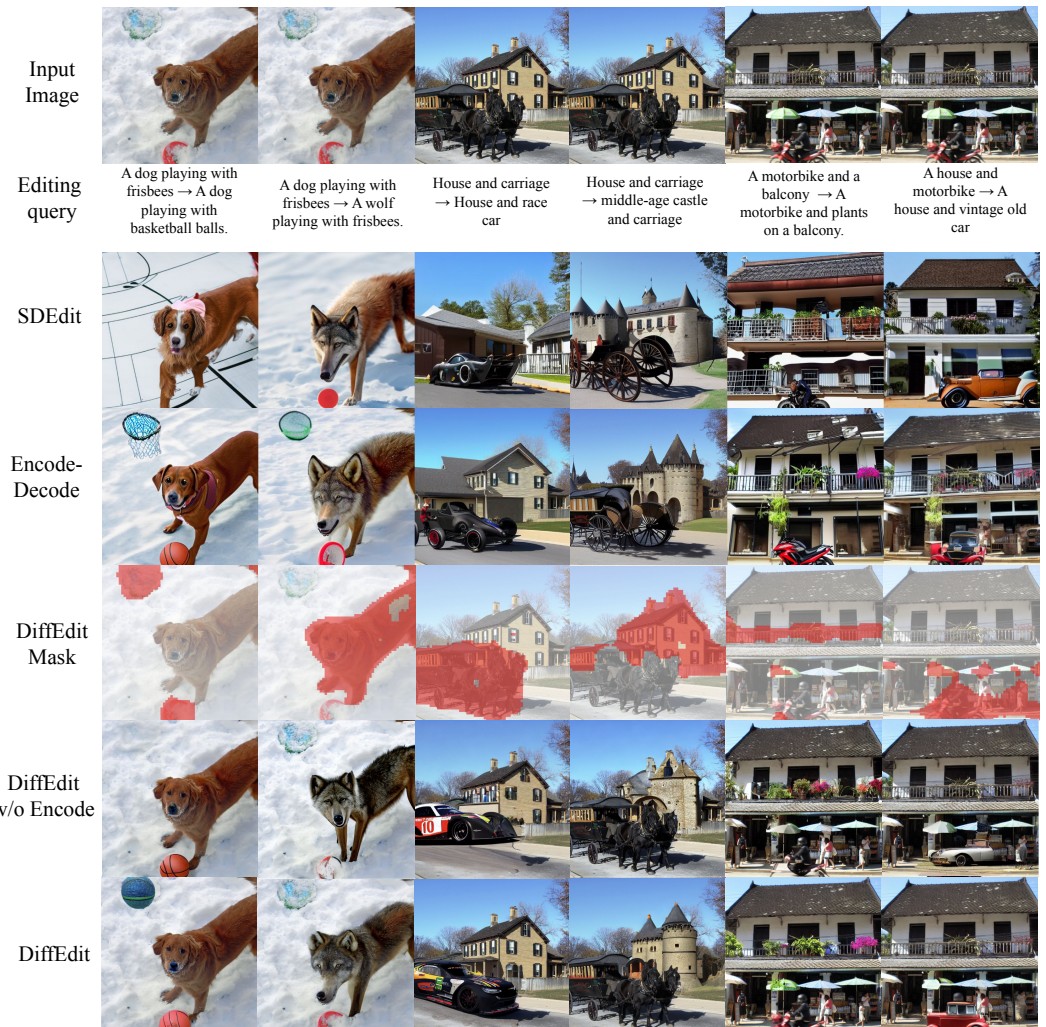

Figure 19: More qualitative examples on COCO. In the first column, the color of the objects to be edited is maintained, which would not be the case with regular inpainting methods. Contrasting similar text query and reference text allows to select the object to be edited.

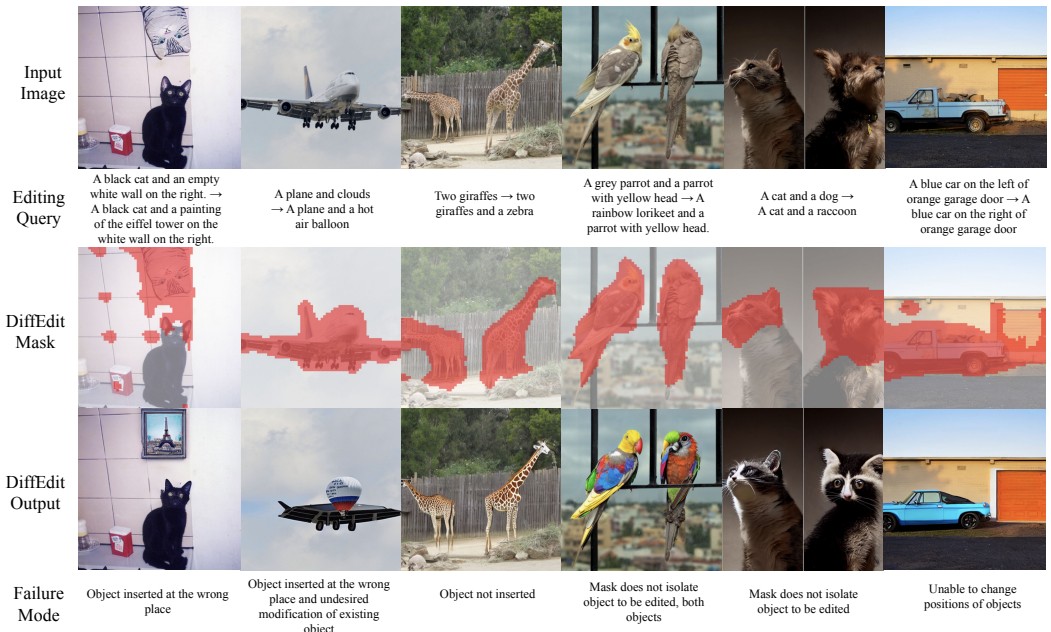

Figure 20: Illustration of failure modes. In the first two columns show difficulty to insert an object in a smooth region of the image. In column three the mask fails to identify a region where to add the zebra. Columns 4 and 5 show mask identification errors, where multiple similar objects are included in the mask, whereas matching the text query only requires to edit a single object. In both cases this results in over-editing. Col. 6 shows the failure to change a spatial relation in the image.

# B    THEORETICAL RESULTS

Here, we prove the bounds given in the main paper. We reused notations from Proposition 1 in the main paper. We also discuss links to optimal transport.

## B.1    PROOF OF SDEDIT BOUND

**Proposition 2.** *Suppose that* $\|\epsilon_\theta(\mathbf{x}, Q, t)\|_2 \leq C$ *for all* $x \in \mathcal{X}$, $t \in [0, 1]$. *Then*

$$\mathop{\mathbb{E}}_{\substack{(\mathbf{x}_0, Q) \sim p_D \\ \epsilon \sim \mathcal{N}(0,1)}} \|\mathbf{x}_0 - D_r(G_r(\mathbf{x}_0, \epsilon), Q)\|_2 \leq (C + 1)\tau \tag{6}$$

*Proof.* Let $T$, $\mathbf{x}_r = G_r(\mathbf{x}_0, \epsilon)$, $\mathbf{y}_r = \mathbf{x}_r$ and $\mathbf{y}_0 = D_r(\mathbf{y}_r, Q)$. Then

$$\|\frac{\mathbf{x}_r}{\sqrt{\alpha_r}} - \mathbf{y}_0\| = \|\frac{\mathbf{y}_r}{\sqrt{\alpha_r}} - \frac{\mathbf{y}_0}{\sqrt{\alpha_0}}\| = \|\int_\tau^0 \epsilon_\theta(x_t, Q, t) d\tau\| \leq C\tau. \tag{7}$$

Since $\frac{\mathbf{x}_r}{\sqrt{\alpha_r}} = \mathbf{x}_0 + \tau\epsilon$, we have $\|\mathbf{x}_0 - \mathbf{y}_0\| \leq \|\mathbf{x}_0 + \tau\epsilon - \mathbf{y}_0\| + \|\tau\epsilon\| \leq C\tau + \tau$ which concludes the proof. $\qquad\square$

In the SDEdit paper (Meng et al., 2021), a proof similar to what we state is given, with three main differences: (i) the proof is given in the case of variance-exploding Stochastic Differential Equation (VE-SDE), which needs adaption for our setting which uses variance-preserving SDE; (ii) the bound is derived in the case of a stochastic differential equation, whereas we use a deterministic DDIM process; (iii) the bound is given by controlling the probability tail, whereas we only consider the expectancy of edit distance. However, despite these differences, the spirit of the proof is the same as here.

## B.2    PROOF OF PROPOSITION 2

**Proposition 3.** *Suppose that* $\epsilon_\theta(\cdot, Q, t)$ *is* $K_1$-*lipschitz and* $\kappa_2$ *defined as*

$$\kappa_2(\mathbf{x}_0) = \max_{t \in [0,1]} \|\epsilon_\theta(E_t(\mathbf{x}_0), Q, t) - \epsilon_\theta(E_t(\mathbf{x}_0), \emptyset, t)\| \tag{8}$$

*Let* $K_2 = \mathbb{E}_{\mathbf{x}_0} \kappa_2(\mathbf{x}_0)$. *Then for all encoding ratio* $r$, *with* $\tau = \sqrt{\alpha_r^{-1} - 1}$,

$$\mathbb{E}_{\mathbf{x}_0} \|\mathbf{x}_0 - D_r(E_r(\mathbf{x}_0), Q)\| \leq \frac{K_2 \tau}{\sqrt{\tau^2 + 1}} \left(\tau + \sqrt{\tau^2 + 1}\right)^{K_1} \tag{9}$$

*Proof.* Let $\sigma$ be a time-dependent variable defined as $\sigma(t) = \sqrt{\alpha_t^{-1} - 1}$. Let $\mathbf{u} = \mathbf{x}/\sqrt{\alpha} = \mathbf{x}\sqrt{1 + \sigma^2}$ and $\mathbf{v} = \mathbf{y}\sqrt{1 + \sigma^2}$. $\mathbf{u}$ and $\mathbf{v}$ are solutions of the following differential system:

$$d\mathbf{u}|_t = \epsilon_\theta(\mathbf{u}/\sqrt{1 + \sigma^2}, \emptyset, t) d\sigma, \tag{10}$$

$$d\mathbf{v}|_t = \epsilon_\theta(\mathbf{v}/\sqrt{1 + \sigma^2}, Q, t) d\sigma, \tag{11}$$

$$\mathbf{u}(r) = \mathbf{v}(r) = E_r(\mathbf{x}_0)\sqrt{1 + \sigma^2}. \tag{12}$$

Let $\mathbf{w} = \|\mathbf{u} - \mathbf{v}\|$, then $\mathbf{w}|_{t=r} = 0$ and

$$d\mathbf{w}|_t \leq \|d\mathbf{u}|_t - d\mathbf{v}|_t\| = \|(\epsilon_\theta(\mathbf{x}, \emptyset, t) - \epsilon_\theta(\mathbf{y}, Q, t)) d\sigma\| \tag{13}$$

$$\leq \|(\epsilon_\theta(\mathbf{x}, \emptyset, t) - \epsilon_\theta(\mathbf{x}, Q, t)\| d\sigma + \|(\epsilon_\theta(\mathbf{x}, Q, t) - \epsilon_\theta(\mathbf{y}, Q, t)\| d\sigma \tag{14}$$

$$\leq \kappa_2(\mathbf{x}_0) d\sigma + K_1 \|\mathbf{x} - \mathbf{y}\| d\sigma \tag{15}$$

$$\leq \left(\kappa_2(\mathbf{x}_0) + \frac{K_1}{\sqrt{1 + \sigma^2}} \mathbf{w}\right) d\sigma. \tag{16}$$

By integration we get

$$\boldsymbol{w}(t) \leq \kappa_2(\mathbf{x}_0) * (\tau - t) + \int_t^\tau \frac{K_1}{\sqrt{1 + \sigma^2}} \boldsymbol{w}(\sigma) d\sigma.$$

From here we can apply Grönwall's inequality:

$$\boldsymbol{w}(0) \leq \kappa_2(\mathbf{x}_0)\tau \exp\left(\int_0^\tau \frac{K_1}{\sqrt{1 + s^2}} ds\right) \tag{17}$$

$$\leq \kappa_2(\mathbf{x}_0)\tau \exp\left(K_1 \log(\tau + \sqrt{\tau^2 + 1})\right) \tag{18}$$

$$\leq \kappa_2(\mathbf{x}_0)\tau \left(\tau + \sqrt{\tau^2 + 1}\right)^{K_1}. \tag{19}$$

Which finally gives

$$\|\mathbf{x}_0 - \mathbf{y}_0\| \leq \frac{\kappa_2(\mathbf{x}_0)\tau}{\sqrt{\tau^2 + 1}} \left(\tau + \sqrt{\tau^2 + 1}\right)^{K_1}. \tag{20}$$

Taking the expectation w.r.t. the input image $\mathbf{x}_0$ gives the final result:

$$\mathbb{E}_{\mathbf{x}_0} \|\mathbf{x}_0 - D_T(E_T(\mathbf{x}_0), Q)\| \leq \frac{K_2 \tau}{\sqrt{\tau^2 + 1}} \left(\tau + \sqrt{\tau^2 + 1}\right)^{K_1} \tag{21}$$

which concludes the proof.

$\square$

### B.3 LINKS TO OPTIMAL TRANSPORT THEORY

The reverse DDIM encoder $E_r$ maps the distribution of images $p_0 = p_D$ to the distribution $p_r$ of images noised at timestep $r$. Khrulkov & Oseledets (2022) suggested that $E_r$ could be an optimal transport map between $p_0$ and $p_r$, minimizing the transport cost $\mathbb{E}_{\mathbf{x}_0} \|\mathbf{x}_0 - E_r(\mathbf{x}_0)\|_2^2$. This means that the encoded images are, on average, as close as possible to the input images, while following the correct distribution $p_r$. It would entail that the unconditional decoder $D_r = E_r^{-1}$ would be an optimal transport map between $p_r$ and $p_0$, and moreover that the conditional decoder $D_r(\cdot, Q)$ would be an optimal transport map between the distributions $p_r(\cdot|Q)$ and $p_0(\cdot|Q)$ conditioned by text description $Q$. Under the hypothesis that $p_r$ is very close to $p_r(\cdot|Q)$, then the Encode-Decode algorithm would be the combination of two optimal transport maps $E_r$ and $D_r(\cdot, Q)$, mapping $p_0$ to $p_r$ and then $p_r \simeq p_r(\cdot|Q)$ to $p_0(\cdot|Q)$. This is a very interesting property and we make the connection with the desired properties of semantic image editing, which can be expressed as an optimal transport problem. Given two distribution of images $p_1, p_2$ (lets say *cats* and *dogs*), the aim is to find the function $f$ that performs the expected edit (changing images of *cats* into images of *dogs*) while minimally editing the image, which can be expressed mathematically as:

$$f = \arg\min_f \mathbb{E}_{\mathbf{x}} \|\mathbf{x} - f(\mathbf{x})\| \quad \text{s.t.} \quad p_2 = f_\# p_1, \tag{22}$$

where $f_\#$ is the push-forward measure. The function $D_r(\cdot, Q) \circ E_r$ is not a solution of this optimal transport problem, because (i) it was proven that the reverse DDIM encoder is not the optimal transport map for some distributions (Lavenant & Santambrogio, 2022), and (ii) the composition of two optimal transport maps is not necessarily an optimal transport map. However, experiments and numerical simulations suggest that $E_r$ is very close from an optimal transport map. It would be interesting to study the "optimality defect" of $E_r$ and of the editing function $D_r(\cdot, Q) \circ E_r$. We leave this for future work.

