# OpenReview forum: "DiffEdit: Diffusion-based semantic image editing with mask guidance"
_ICLR.cc/2023/Conference — ICLR 2023 notable top 25%_

### Official Review · Reviewer_hxgu · 2022-10-24

**Confidence:** 4
**Correctness:** 4
**Technical Novelty And Significance:** 3
**Empirical Novelty And Significance:** 3
**Recommendation:** 8

**Clarity, Quality, Novelty And Reproducibility:**

This paper is clearly written and well organised. Given the description I think it is reproducible. While the current description in Sec 3.2 Step 1 is acceptable, I would prefer to use an equation to describe how the mask is obtained. Besides, the meaning of Dr in Eq(4)(5) is not explained. I think it’s the DDIM decoding process, but it should be mentioned.
The method is simple but is new. In general, the quality of the paper looks good to me.

**Strength And Weaknesses:**

Strength:

- The paper proposes a new method to automatically estimate the mask that needs to be edited, which reduces users' effort. The method of contrasting the predictions conditioned on different texts looks reasonable to me.
- DIFFEDIT proposes to use the reverse DDIM step to encode the image, which is shown to lead to more consistent image editing results.
- Combining the proposed two techniques, the method achieves better image editing than previous methods both qualitatively and quantitatively on several datasets.

Weaknesses:

- For the COCO dataset, only qualitative results of the proposed method are shown. It is recommended to add a qualitative comparison with other baselines in the Appendix.
- For the current method, the predicted mask is much more reasonable when a reference text is provided. However, in the application of real image editing, the reference text is often not available, so a minor issue is that the author needs to provide both query text and reference text for the method to perform better.

**Summary Of The Paper:**

This paper proposes DIFFEDIT, a new method for image editing based on diffusion models. The main contribution is a method to automatically discover the areas that need to be edited by contrasting the predictions conditioned on different texts. Besides, the paper shows that using the latent inference capability of DDIM achieves more consistent editing. Experiments on ImageNet, COCO, and an Imagen generated dataset demonstrate that DIFFEDIT achieves better image editing than previous baselines.

**Summary Of The Review:**

While I feel the proposed method is pretty simple, it is reasonable and effective in producing region-specific and consistent image editing as proved in experiments. Object specific image editing is an important problem and I think this work would be quite useful. A minor weakness in qualitative evaluation could be further improved. To summarise, I tend to accept this paper.

---

> ### Author Response · Authors · 2022-11-10
> **Answer to Reviewer hxgu**
>
> Thank you for your time and effort reviewing our paper, as well as very positive feedback. We will update the paper with recommended additions. As you mention, to get the most out of DiffEdit, a reference text should be provided. We think that this can be the basis of future work: finding the reference text automatically when the user only provides the input image and text query, which interestingly corresponds to solving the same problem as in our paper with modalities switched: how to modify a text caption (here, an editing query) as little as possible to match the input image as much as possible?

---

### Official Review · Reviewer_zQQb · 2022-10-25

**Confidence:** 4
**Correctness:** 4
**Technical Novelty And Significance:** 1
**Empirical Novelty And Significance:** 2
**Recommendation:** 5

**Clarity, Quality, Novelty And Reproducibility:**

The paper is well written and easy to understand,

The reproducibility should be possible given the simplicity of the method

**Strength And Weaknesses:**

Strength
 - theoretical analysis is nice to have
 - the generated mask seems reasonable
 - the method is simple. Auto generate the mask from 2 version of slightly different texts then encode/decode to the appropriate image.
 - experiments are on (relatively) large scale datasets

Weakness
 - The use case is not quite clear to me. The mask generation from the different in text query and reference text means that you need to have both. So from what I understand, this is only applicable for image editing on an existing text-to-image pipeline? (Also Isn’t that more like giving more control over image generation than an image editing?)
 - I think the simplicity of the method is both the strength and the weakness. It is good that it is simple, so more people can use it. But it also mean there isn’t much insight/impact gain from reading the paper.
- From the result figure, it is not an obvious improvement to me among each baselines. Using mask does help with keeping the background the same as the original query, but that seems to be the only benefit of the proposed method.

**Summary Of The Paper:**

The authors propose a 3 steps image editing pipeline that 1) generate segmentation mask from the change in text query, 2) encode the image with diffusion process until a time step ‘r’, and 3) decode it back to the image via reverse diffusion process condition on the text query, and with mask as the guidance. They show results on ImageNet and coco dataset, as well as image generated from imagen.

**Summary Of The Review:**

Unfortunately, I don’t think I currently see enough contributions to recommend acceptance. Given the weakness I mentioned, I think it is slightly below the bar for acceptance.

---

> ### Author Response · Authors · 2022-11-10
> **Answer to Reviewer zQQb**
>
> Thank you for your time reviewing our paper.
>
> Q: Method only applicable for image editing on an existing text-to-image pipeline
> A: We emphasize that this is rather a strength: DiffEdit can leverage a diffusion model trained for image generation only, and adapt it without any additional training cost to an image editing task. This avoids editing-specific training, which is computationally expensive, especially given that a lot of knowledge can be extracted from text-conditioned image generation models like Stable Diffusion.
>
> Q: you need to have both text query and reference text
> Providing a reference text describing the image is optional in our method, but can help to produce better masks for complex text prompts. In the paper, we experiment with both the case when we have access to the reference text and the case where it is unavailable. On ImageNet, results are presented without reference text for fair comparison with other methods. Figures 7 and 10 show a quantitative analysis for both cases, demonstrating good performance even without reference text. Since the performance is still better when a reference text is available, this can motivate future work to tackle the following problem: How to find the reference text automatically given the input image and text query only?
>
> Q: insights
> As mentioned above, we provide insights in the paper on the difference between using or not a reference text. Besides, we provide a theoretical analysis shedding light on why DDIM encoding helps to better edit images. Qualitative examples allow to validate that the mask indeed covers the image region that should be edited.
>
> Q:benefits of proposed method
> Quantitative analysis on ImageNet, Imagen-dataset and COCO demonstrate that DiffEdit performs more better edits that other methods: better alignment with text query (CLIPScore) with overall less image modifications (LPIPS). This is due to two factors: first, DiffEdit identifies what must be changed in an image, which corresponds to the background on ImageNet but to other image parts in the other datasets  (e.g. not modifying central object figure 9 col 4). Second, the DDIM encoding component, on top of the mask-aware decoding process, allows to improve quantitative results (Ablation Figure 6.a), because it maintains properties of the input image as we can see in qualitative examples Figure 5. Specifically, columns 1, 4, 7, 8, 9 (Figure 5) show that the pose of objects is better maintained.

---

### Official Review · Reviewer_ZwKd · 2022-10-27

**Confidence:** 5
**Correctness:** 4
**Technical Novelty And Significance:** 2
**Empirical Novelty And Significance:** 3
**Recommendation:** 8

**Clarity, Quality, Novelty And Reproducibility:**

Clarity: I think the paper is very clear.

Quality: The results are of high quality. The extensive evaluation shows the state-of-the-art performance when compared to many recent baselines.

Novelty: As discussed in the weakness section, I think the weakest part of the paper is that both spatial masking for editing and DDIM encoding are not intrinsically novel.

Reproducibility: The paper builds on publicly available models and data. The paper also includes sufficient implementation details. I believe that reproducibility would not be an issue.

**Details Of Ethics Concerns:**

Image editing applications naturally have potential harmful use (e.g., creating fake news). The paper discussed in the ethics statement that they will release the code under a license similar to the Stable Diffusion.

**Strength And Weaknesses:**

Strength:
+ The paper is very well-written. The exposition is clear. The figures are informative.
+ The proposed approach is technically sound.
+ The evaluation of three datasets (ImageNet, images generated from Imagen, and COCO) is solid and convincing. Both the quantitative and qualitative results demonstrate the effectiveness of the method.
+ The ethics statement and potential misuse of the techniques are thoroughly discussed.

Weakness:
- The concepts of predicting spatial masks for semantic image editing have been extensively explored in other image-to-image translation literature. Examples include:
[CVPR 2018] Da-gan: Instance-level image translation by deep attention generative adversarial networks
[NeurIPS 2018] Unsupervised Attention-guided Image-to-Image Translation
[ICLR 2019] InstaGAN: Instance-aware Image-to-Image Translation
The difference here is that the generative models at the time were GANs instead of diffusion models.

- Another core difference is the use of DDIM encoding as opposed to directly adding noises in the original image. DiffusionCLIP [CVPR 2021] also uses the DDIM step to encode the input image. The key difference is that the proposed method relies on a pre-trained text-2-image diffusion model as opposed to CLIP used in DiffusionCLIP. This allows us to avoid the costly model fine-tuning.

**Summary Of The Paper:**

The paper presents a method for performing text-based semantic image editing. The key ideas are
1) identifying where to edit by comparing the image differences between query text-guided and reference text-guided (or unguided) image generation and
2) use DDIM encoding for preserving the contents that are outside the generated masks.

For 1), compared to the existing semantic editing method, the proposed mask generation automates the step of manually specifying the mask for editing.
For 2), the DDIM encoding preserves the background content more faithfully when compared with current practices (that add noise to the original image as in SDEdit).

The method can achieve semantic editing while maintaining the background contents in the original image.
The paper evaluates the method in three settings, ImageNet, images generated from Imagen, and COCO. The experimental results validate the capability of the proposed method for image editing.

**Summary Of The Review:**

The paper combines the ideas of spatial masking and DDIM encoding to improve the quality of semantic image editing. None of the techniques are new, but it appears to be an interesting combination in this application context. The paper shows strong evaluation results on three datasets and reports convincing quantitative and visual results. I think the community would benefit from this simple approach for semantic image editing.

---

> ### Author Response · Authors · 2022-11-10
> **Answer to Reviewer ZwKd**
>
> First, thank you for your time reviewing our paper. We will add mentioned references in the paper.
>
> Q: novelty of spatial masking
> While using spatial masking has been studied in the literature, we would like to highlight that it is our technical solution to automatically finding an editing mask that is novel. Besides the difference in base architecture (diffusion models vs GANs), the DiffEdit mask algorithm is training-free and only leverages a pre-trained text-to-image diffusion model (also used in other components of the method). In contrast:
> - InstaGAN takes as additional input object segmentation masks without computing them, whereas we provide a mask detection algorithm that can segment objects as well as more generally find relevant regions to edit.
> - In DA-GAN and Unsupervised Attention-guided Image-to-Image Translation, masks are computed with a network trained with the entire architecture.

---

### Official Review · Reviewer_WyT5 · 2022-10-28

**Confidence:** 4
**Correctness:** 4
**Technical Novelty And Significance:** 4
**Empirical Novelty And Significance:** 3
**Recommendation:** 10

**Clarity, Quality, Novelty And Reproducibility:**

Clarity:

Very well written. Just by looking ant fig 2 the reader understands what is happening inside a minute.

Quality:

Very high.

Novelty:

Novel to me.

Reproducibility:

Just by reading the paper one can easily reproduce the method, provided one has the access to the trained models.


**Strength And Weaknesses:**

Strengths:

The process is extremely simple:
- step1: estimate the mask (that a user otherwise would provide)
- step2: run DDIM encoding
- step3: run DDIM decoding conditioned on query text, and keep pixel values outside the mask

The key idea is very intuitive:
"When the denoising an image, a text-conditioned diffusion model will yield different noise estimates given different text conditionings. We can consider where the estimates are different, which gives information about what image regions are concerned by the change in conditioning text."

The paper is very well written.

The solution is supported by theoretical analysis, where they show bounds on how far the edited images are.

Ablations show the main effect on design choices (fig 5, fig 6) and quantitativelly on the main hyperparameters (fig 6 binarisation threshold and fig 4 encoding ratio).

The method is extensively evaluated on different datasets and against other methods, and it beats the SOTA (fig 4)

Weaknesses:

I did not find any

**Summary Of The Paper:**

The paper proposes a novel method for text based image editing, that addresses the shortcomings from previous methods:
- either a mask had to be provided by the user
- or the edits modified the background

In this paper the user does not have to provide a mask and the background will not change either. They achieve that by estimating the mask instead of requiring the user to provide it.


**Summary Of The Review:**

The paper have strengths in many areas. Unless I missed something, it is a clear accept.

---

> ### Author Response · Authors · 2022-11-10
> **Answer to Reviewer WyT5**
>
> Thank you for reviewing our paper and for the very positive feedback, pointing out that the paper has “strength in many areas” !

---

### Author Response · Authors · 2022-11-10
**General comment to reviewers**

We would like to thank the reviewers for their efforts, detailed reviews and interest for our submission. We will integrate all their remarks in the revised version of the paper that we will post next week.

---

### Decision · Program_Chairs · 2023-01-20

**Decision:**

Accept: notable-top-25%

**Justification For Why Not Higher Score:**

This is a nice and simple algorithm for image editing. If this is CVPR, I would recommend the paper for oral. But as this is ICLR, which emphasizes machine learning more, I would recommend spotlight as this is not a very general machine learning method.

**Justification For Why Not Lower Score:**

The paper is clearly better than other poster papers in terms of presentation and potential impact. I would argue not to lower it to a poster.

**Metareview: Summary, Strengths And Weaknesses:**

The paper proposes a method to edit a real image. Given an input real image, a text description of the input image, and a text description of the target output image, the proposed method first estimates the foreground object mask assuming the difference between the source and target text descriptions is only the foreground object. It then encodes the input image to the noise space and then decodes it with the target text description, where the estimated mask is used to preserve the background via the replacement method.

The paper receives 4 reviews. Three reviewers are very happy with the paper. They think the presentation is clear, the theoretical analysis is nice, and the presented performance is convincing. One reviewer rated the paper slightly below the bar but did not provide convincing arguments why this is the case. In general, a simple method that does the job is preferred.

Based on the supportive reviews, the effectiveness of the proposed method, and its potential use in various image editing applications, the AC would like to recommend acceptance of the paper.

**Note From Pc:**

if the above contains the word "oral" or "spotlight" please see: "oral" presentation means -> notable-top-5% and "spotlight" means -> notable-top-25%. As stated in our emails, we are disassociating presentation type from AC recommendations